# Topical Delivery of Ketorolac Tromethamine via Cataplasm for Inflammatory Pain Therapy

**DOI:** 10.3390/pharmaceutics15051405

**Published:** 2023-05-04

**Authors:** Zhiyuan Hou, Qiang Wen, Wenhu Zhou, Peng Yan, Hailong Zhang, Jinsong Ding

**Affiliations:** 1Xiangya School of Pharmaceutical Science, Central South University, Changsha 410006, China; 2Changsha Jingyi Pharmaceutical Technology Co., Ltd., Changsha 410006, China

**Keywords:** ketorolac tromethamine, topical drug delivery system, cataplasm, anti-inflammatory, analgesic

## Abstract

Nonsteroidal anti-inflammatory drugs (NSAIDs) have been widely used in the treatment of inflammatory pain, such as in osteoarthritis. Ketorolac tromethamine is considered to be an NSAID with strong anti-inflammatory and analgesic potency, however, traditional applications, such as oral administration and injections, often induce high systemic exposure, leading to adverse events such as gastric ulceration and bleeding. To address this key limitation, herein we designed and fabricated a topical delivery system for ketorolac tromethamine via cataplasm, which is based on a three-dimensional mesh structure formed by the cross-linking of dihydroxyaluminum aminoacetate (DAAA) and sodium polyacrylate. The viscoelasticity of the cataplasm was characterized by rheological methods and exhibited a “gel-like” elastic property. The release behavior showed a Higuchi model characteristic with a dose dependence. To enhance the skin permeation, permeation enhancers were added and screened utilizing ex vivo pig skin, in which 1,2-propanediol was found to have the optimal permeation-promoting effect. The cataplasm was further applied to a rat carrageenan-induced inflammatory pain model, which showed comparable anti-inflammatory and analgesic effects with oral administration. Finally, the biosafety of the cataplasm was tested in healthy human volunteers, and reduced side effects were achieved as compared to the tablet formulation, which can be ascribed to less systemic drug exposure and lower blood drug concentrations. Therefore, the constructed cataplasm can reduce the risk of adverse events while maintaining efficacy, thus serving as a better alternative for the treatment of inflammatory pain, including osteoarthritis.

## 1. Introduction

Inflammatory pain, which is initiated by trauma, infection, chemical stimulation, and surgery, is a type of common disease. Osteoarthritis is one inflammatory pain disease with a very high incidence. It is estimated that more than 240 million people suffer from osteoarthritis worldwide, and the number of people in the United States alone exceeds 54 million [1,2]. The clinical characteristics of osteoarthritis can be persistent or intermittent, which brings great distress to patients [2,3]. Therefore, choosing an appropriate treatment for pain relief is of paramount importance.

Nonsteroidal anti-inflammatory drugs (NSAIDs) are widely used as a first-line treatment for osteoarthritis. As one of them, ketorolac tromethamine can reduce the production of prostaglandins (PG) by inhibiting the activity of cyclooxygenase (COX), thereby producing anti-inflammatory and analgesic effects. More importantly, ketorolac tromethamine can achieve an analgesic effect similar to that of opioids but without addiction [2,4,5,6]. Although effective, current dosage forms (e.g., injections and tablets) of ketorolac tromethamine result in systemic drug exposure, especially at noninflammatory sites, which may lead to adverse effects, such as gastric ulcers [7,8].

Therefore, it is greatly important to reduce the occurrence of adverse effects of ketorolac tromethamine while preserving the anti-inflammatory and analgesic efficacy. The development of ketorolac tromethamine eye drops (ACUVAIL^®^) was based on this concept and are indicated for the treatment of pain and inflammation following cataract surgery [9,10,11]. Considering the characteristics of inflammatory pain [12], the delivery of ketorolac tromethamine through the skin to achieve local drug enrichment in the pain site is expected to accomplish this clinical need. Yang et al. [13] and Moussaoui et al. [14] have developed gels loaded with ketorolac tromethamine for periodontal pain and the postoperative treatment of acromegaly, respectively. Furthermore, the gel systems based on liposomal and nanodelivery technologies have shown more effective transdermal drug penetration and therapeutic effects [15,16]. However, when applied to the skin, the gel form has its own drawbacks, such as easy erasability, short duration of efficacy, and low patient compliance.

Here, based on the cross-linking reaction between dihydroxyaluminum aminoacetate (DAAA) and sodium polyacrylate without organic solvents, a three-dimensional reticulated skeleton of cataplasm was successfully constructed to deliver ketorolac tromethamine topically for the first time. Its matrix is believed to achieve a longer and constant rate of sustained drug release [17], which can be particularly useful for providing pain relief over an extended period. Furthermore, cataplasm can also be used to provide a cooling effect to the applied area, which can help to reduce pain, inflammation, and swelling. The constructed cataplasm has the advantages of low skin irritation, controllable preparation process, and good reproducibility, and is suitable for industrial production. In this work, the cross-linking and molding mechanisms of the cataplasm are systematically characterized and discussed in detail, along with the adhesion, rheological, in vitro release, and permeation characteristics. The cataplasm developed herein has demonstrated excellent efficacy and safety in animal pharmacodynamic and human pharmacokinetic studies and holds promise for commercial transfer, hopefully providing a new and effective option for patients in the future. 

## 2. Materials and Methods

### 2.1. Animals

The SD rats (200 ± 20 g) were purchased from Tianqin Biology (Changsha, China), whose quality certificate number is 430726211100339273 and license number is SYXK (Xiang) 2020-0015. To exclude the influence of animal gender, half of the purchased SD rats were male and half were female. The acquisition of and experiments on the SD rats were approved by the ethics committee of Xiangya School of Pharmaceutical Science, Central South University.

### 2.2. Materials

Ketorolac tromethamine was purchased from Renan Pharmaceutical (Chengdu, China). Commercially available ketorolac tromethamine tablets were manufactured by Mylan Pharmaceutical (Pittsburgh, PA, USA). Isopropyl myristate, 1,2-propanediol, and tartaric acid were obtained from Sigma-Aldrich (St. Louis, MO, USA). Polyglycerol-3 oleate (Plurol^®^ Oleique CC497) and diethylene glycol monoethyl ether (Transcutol^®^) were gifted by Gattefossé (Saint-Priest, Rhone, France). Partially neutralized polyacrylate (ViscomateTM, NP700) was obtained from Showa Denko KK (Kawasaki, Kanagawa, Japan). DAAA was purchased from Xiyue Pharmaceutical (Weinan, China). *λ*-carrageenan was received from Ika Bio (Shanghai, China). The artificial membrane (With 0.45 μm aperture, polyethersulfone material) was obtained from Keelong Laboratory Equipment (Tianjin, China). The isolated skin of Bama minipigs was purchased from Jingde Agricultural Products (Xingtai, China).

### 2.3. Preparation of Ketorolac Tromethamine Cataplasm

Briefly, dibutyl hydroxytoluene (BHT), DAAA, kaolin, disodium edetate (EDTA-2Na), titanium dioxide, partially neutralized polyacrylate, and povidone K90 were dispersed uniformly in polyethylene glycol (PEG) 400 with gentle mechanical stirring, which was named phase A. Sorbitol, tartaric acid, and 20% polyacrylic acid solution were sequentially dissolved in purified water with mechanical stirring, which was called phase B. Ketorolac tromethamine was completely dissolved in purified water with mechanical stirring, which was called phase C. Menthol, methylparaben, and propylparaben were added to 1,2-propanediol and mechanically stirred until they dissolved, which was called phase D.

Phase D was added to phase A with mechanical stirring for 30 min. Then, phase C was added to the mixture of phase A and phase D. Phase B was poured into a stirring kneader (Shenglong Chemical Machinery, Laizhou, China), and then the mixture of phase A, phase C, and phase D was added while kneading. After kneading evenly, the material was coated, laminated, and cut with a cataplasm coating machine (Xinyi Huida Electromechanical Equipment, HD-SIII type, Beijing, China). The cut samples were cross-linked at room temperature for seven days to form the cataplasm. The obtained ketorolac tromethamine cataplasm was sealed and stored at room temperature.

### 2.4. Microstructure Observation

#### 2.4.1. Polarized Light Microscopy Characterization

Ketorolac tromethamine was uniformly dispersed in light liquid paraffin and observed using a polarizing microscope (Motic China Group, Panthera TEC-POL, Xiamen, China). A small amount of paste on the prepared cataplasm was scraped off and observed using the above-mentioned microscope.

#### 2.4.2. ESEM Characterization

The test was carried out using an environmental vacuum scanning electron microscope (Thermo Scientific, Quattro, Waltham, MA, USA). A small amount of cataplasm paste was scraped and placed in purified water to swell for 6 min and then frozen using liquid nitrogen. The sample was placed on the cold table, and the surface morphology of the sample was observed. Test conditions: low-vacuum mode was selected, voltage was 7 kV, signal source was GSED, test pressure was maintained in the range of 50 to 700 Pa, cold table temperature was in gradient heating mode, temperature range was −20 °C to 10 °C, and heating rate was 10 °C/min.

### 2.5. Tack Test

There were two methods to detect the tack of ketorolac tromethamine cataplasm.

The first method was called the rolling-ball tack test [18], and the tack was measured using a SUS304 inclined ball tack test device (Sumspring Co., Ltd., Jinan, China). First, the cataplasm was cut into a rectangular shape of 5.0 cm * 2.5 cm. Second, the backing membrane side of the cataplasm was fixed on a flat plate with an inclination angle of 15°, and the release liner was removed to expose the paste to the air. The rolling length of the ball on the cataplasm was 5.0 cm, and the distance between the initial position of the ball and the upper end of the test sample was 1.0 cm. Finally, the ball fell freely along the inclined plane and the largest ball that could stick to the cataplasm had its number recorded. Parallel operations were performed three times, and the largest ball number among the three results was recorded.

The second method was called the loop tack test [19]. The cataplasm was cut into a rectangular shape of 2.4 cm * 10.0 cm, then the short sides were bent and glued together to form a ring. Next, the sample was clamped to the loop tack test device (Saicheng Electronic Technology Co., Ltd., Jinan, China) so that the annular bottom of the sample was 30 mm from the top of the test plate. Then, the sample was slowly lowered 45 mm so that the paste side of the annular sample was in contact with the standard steel test plate. After 1 min contact time, the sample was pulled up at a rate of 100 mm/min, and the maximum force to separate the sample from the standard steel test plate was recorded. The average value of three parallel experiments was taken.

### 2.6. Peel-Strength Test

PSTC 1 standard test method evaluated peel strength at a peel angle of 180° using an electronic peel-strength test device (Sumspring Co., Ltd., Jinan, China) [20]. The backing membrane side of the cataplasm was fixed on a stainless-steel plate (length: 125 mm, width: 50 mm, and thickness: 2 mm). Then, the release liner was removed, and a polyester film covered the paste. After that, a pressing roller weighing 2 kg was rolled back and forth on the polyester film three times, and the sample was placed at room temperature for 30 min to ensure that the polyester film and the paste were fully contacted and bonded. The polyester film was driven with the clamp to move at a constant speed of 300 mm/min, and the average force required to peel the polyester film from the paste was recorded. The average value of three parallel experiments was taken.

### 2.7. Rheological Test

The rheological properties of cataplasm (dose of 1.0%) were studied using a rheometer (TA Instrument, DHR-1, New Castle, DE, USA) and tests were performed using a 25 mm flat plate certificate. Due to the high elasticity of the cataplasm, an axial-force-adjustment step was added before each test (axial force: 1 N; sensitivity: 0.1 N). Amplitude scans and frequency scans of the samples were performed in oscillation mode. Creep-recovery tests were performed in step (transient)-creep mode. Viscosity profile scans were performed in flow-ramp mode.

#### 2.7.1. Oscillation Amplitude

The sample was exposed to increasing stresses from 10 to 3000 Pa at 32 °C with a constant 1 Hz frequency. Then, the storage modulus (G′), loss modulus (G″), and loss tangent (tan δ) values were plotted with a logarithmic scale. This oscillation test reflected the linear viscoelastic region (LVR) of the sample, as well as the trend of its viscoelasticity with increasing stress.

#### 2.7.2. Oscillation Frequency

The change in viscoelastic modulus of the sample over a range of frequencies was measured by subjecting the sample to constant stress (50 Pa) at 32 °C [17].

#### 2.7.3. Creep-Recovery Test

The sample was held for 300 s at 32 °C with stress intensity set to 50 Pa. The stress was then immediately relieved, and the recovery was tracked by monitoring the strain over time for 300 s.

#### 2.7.4. Viscosity Curve

The process duration was set to 60 s and the shear rate ranged from 100.0/s to 0.01/s. The variation in sample viscosity and stress with the shear rate was recorded.

### 2.8. In Vitro Release Study

In vitro release behavior of different doses of cataplasm (0.5%, 1.0%, and 1.5%) was studied using the vertical Franz diffusion cell (Yuyan Scientific Instruments, TK-12D, Shanghai, China). Conditions to be met in experiments with the Franz diffusion cell included a 12.0 mL receptor cell, a 1.54 cm^2^ effective diffusion area, 12.0 mL of degassed 0.01 mol/L phosphate buffer receptor liquid (pH 7.4), a diffusion cell temperature of 32 °C, and a 600 rpm stirring speed. The cataplasm was tightly attached to the upper side of the artificial membrane, and 12.0 mL of sample solution was collected from the receptor cell at 0.25, 0.5, 0.75, 1, 2, 4, 6, 8, 10, and 12 h, respectively, and an equal volume of the receptor liquid was added. Samples were filtered through a filter with a pore size of 0.22 μm and analyzed via high-performance liquid chromatography (HPLC). The number of experiments for each dose of cataplasm was six times, and the cumulative release (*Q*, μg/cm^2^) of ketorolac tromethamine was calculated using Equation (1).
(1)$$Q=\frac{\sum_{i=1}^{n}C_{i}*V_{i}}{A}$$
where *C_i_* is the drug concentration at each time point, *V_i_* is the sampling volume at each point, and *A* is the effective diffusion area.

The experimental data before reaching the plateau (0–6 h) were fitted with different mathematical models (including zero-order, first-order, and Higuchi), and the correlation coefficient (r^2^) was used as an indicator to determine the best-fitting model.

### 2.9. Selection of the Type of Skin-Penetration Enhancers

To investigate the difference in the rate and extent of diffusion of ketorolac tromethamine in the skin among different types of transdermal enhancers, an in vitro penetration experiment was performed. Transcutol^®^, 1,2-propanediol, Plurol^®^ Oleique CC497, and isopropyl myristate were used as transdermal penetration enhancers. Referring to other transdermal studies [14], the isolated skin of a 1-month-old Bama minipig was chosen as the permeation barrier because it is similar in structure to human skin.

The skin was thawed at room temperature in 0.9% NaCl solution before application, and residual liquid on the skin’s surface was removed with absorbent paper. Then, clean skin was cut into a circle with a diameter of 3.0 cm and placed on the Franz diffusion cell with a diffusion area of 1.54 cm^2^. The stratum corneum of the skin faced the donor compartment and the dermis was in contact with the receptor fluid (degassed 0.9% NaCl solution) with a volume of 12.0 mL of the receiving cell at 600 rpm in a 32 °C diffusion cell. The cataplasm (cut into circles with a diameter of 1.4 cm), containing different penetration enhancers, adhered to the stratum corneum side of the skin. Samples were collected with a pipette in a volume of 500 μL and supplemented with an equal volume of 0.9% NaCl solution at 1, 2, 4, 8, 12, and 24 h. Samples were analyzed with the HPLC method to calculate the cumulative drug penetration amount per unit area of skin. After sampling, the cataplasm was removed, and the skin was washed with 50% methanol. The skin was then ground into a powder with a freezer grinder (Jingxin Technology, Shanghai, China). Then, methanol was added as a skin extract, sonicated for 10 min, and centrifuged. The supernatant was collected for HPLC analysis to calculate the skin retention of ketorolac tromethamine.

### 2.10. In Vitro Skin Penetration Study

The purpose of this experiment was to study the degree and rate of penetration of ketorolac tromethamine through isolated skin with different doses of cataplasm (0.5%, 1.0%, and 1.5%). The experimental method operated according to item 2.9. The ratio of cumulative-permeation-to-dose at 24 h in each group was calculated. In addition, a linear regression was performed on the cumulative permeation-time curve. The slope of the straight line with a linear correlation coefficient greater than 0.99 was defined as the steady-state transdermal permeation rate per unit area of the drug (*J*), and the intercept of the extrapolation of the regression curve and the time axis was defined as the lag time (*T_lag_*).

### 2.11. Histological Evaluation

#### 2.11.1. H and E Staining

Referring to the method in the literature [21], the isolated skin samples of 1-month-old Bama minipigs, untreated and treated with the cataplasm (doses of 0.5%, 1.0%, and 1.5%, respectively) for 24 h, were repeatedly washed with normal saline, followed by the formulation of 4–7 μm tissue slices. After that, they were stained with hematoxylin and eosin and were observed using a microscope (ZEISS, AxioVert.A1, Oberkochen, Germany).

#### 2.11.2. SEM Characterization

According to the method from previous studies [22,23], the isolated skin of 1-month-old Bama minipigs, untreated and treated with the cataplasm (dose of 1.0%) for 24 h, was fixed with a 2.5% glutaraldehyde solution at 4 °C for 24 h, and the fixative was poured out. After that, the samples were rinsed three times with 0.1 M phosphate buffer (pH 7.0) for 15 min, and the skin was fixed with 1% osmic acid solution for 1–2 h. Then, the osmic acid solution was removed, and the samples were rinsed three times with 0.1 M phosphate buffer (pH 7.0) for 15 min. Finally, the samples were dehydrated, dried, and coated, followed by observation using a scanning electron microscope (Hitachi, SU8010, Tokyo, Japan).

### 2.12. In Vivo Pharmacodynamic Study in SD Rats

#### 2.12.1. Grouping and Administrating

SD rats (10 rats per group) were randomly assigned to the control group, oral group, 0.5% cataplasm group, 1.0% cataplasm group, and 1.5% cataplasm group. In the control group, rats were treated with a cataplasm vehicle. In the oral group, rats were administered using gavage (0.9 mg/kg). In the cataplasm group, rats were treated with cataplasm with different drug contents. After cataplasm administration for 3 h at the right hindfoot or oral administration, rats in all groups were injected subcutaneously with 1% *λ*-carrageenan (0.1 mL) at the right hindfoot pad.

#### 2.12.2. Carrageenan-Induced Paw Edema

The toe volume was measured using a toe volume meter (Yiyan Technology, YLS-7C, Jinan, China) before (0 h) and 0.5, 1, 2, 4, and 6 h after the administration of carrageenan. The degree of paw swelling was calculated as:(2)$$Swelling\;\left(\%\right)=\frac{V_{t}-V}{V}\times 100$$
where, *V_t_* is the volume of the carrageenan-treated paw, and *V* is that of the paw before carrageenan treatment.

#### 2.12.3. Carrageenan-Induced Paw Pain

The pain threshold (PT, gf) of the right hindfoot was measured before (0 h) and 0.5, 1, 2, 4, and 6 h after carrageenan administration, using an electronic Von Frey (Ugo Basile Bioresearch Instruments Inc., e-VF, Gemonio, Italy). The difference in the pain threshold before and after drug administration was calculated. The PT change rate was calculated as follows:(3)$$PT\;Change\;Rate\;\left(\%\right)=\frac{PT_{t}-PT}{PT}\times 100$$
where *PT_t_* is the pain threshold of the carrageenan-treated paw, and *PT* is that of the paw before carrageenan treatment.

### 2.13. In Vivo Pharmacokinetic Study in Humans

#### 2.13.1. Subjects

This trial’s subjects were selected from a healthy Chinese male population. After signing the informed consent form, participants were examined, and screening examinations were completed within 14 days of the study’s start date. Physical examinations, vital signs examinations, 12-lead electrocardiograms, and standard laboratory evaluations (including blood routine, urine, and blood biochemistry) were conducted to screen individuals to ensure that they met the clinical trial requirements. The ethics committee of Xiangya School of Pharmaceutical Science, Central South University, approved the human pharmacokinetic study.

#### 2.13.2. Trial Design

This study was a single-center, open-label, parallel, two-period, two-treatment, fasting, and single-dose clinical trial. Subjects sequentially applied the ketorolac tromethamine cataplasm (one patch at a dose of 1.0%, i.e., 100 mg) to the dorsal skin versus a commercial ketorolac tromethamine oral product tablet (one tablet, 10 mg) during two periods. In the first period (cataplasm-treatment group), about 3 mL of venous blood was collected from subjects before (0 h) and 2, 4, 6, 8, 10, 12, 14, 16, 18, 20, 24, 28, 32, 36, 48, 60, and 72 h after administration in a tube containing sodium heparin. In the second period (tablet-treatment group), blood was collected from subjects before (0 h) and 0.17, 0.25, 0.33, 0.5, 0.75, 1, 1.25, 1.5, 2, 4, 6, 8, 10, 12, 16, 24, and 36 h after administration. The washout period was seven days between the two periods to ensure that the residual drug had been eluted from the subjects’ bodies before the oral administration of the tablets.

#### 2.13.3. Safety and Tolerability Assessments

The investigators constantly examined the physical status of the recruited subjects throughout the study period.

#### 2.13.4. Determination of Drug Concentrations in Human Plasma

Within 60 min of collection, blood samples were placed into a low-temperature centrifuge (GL12, Changsha Yingtai Instruments Co., Ltd., Changsha, China) and centrifuged for 10 min at 2480 g and 4 °C. The samples were placed into an ultra-low-temperature refrigerator at −80 °C for storage and subsequent analysis after centrifugation. The concentration of ketorolac tromethamine in plasma was measured via LC-MS/MS, using agomelatine as an internal standard and plasma from healthy subjects as a biological matrix. The experiment was performed with HPLC (Shimadzu, LC-30AD, Chukyo, Kyoto, Japan) with a C_18_ column (2.1 mm × 50 mm, 1.7 μm) at a flow rate of 0.40 mL/min and a column temperature of 40 °C. The mobile phase was in gradient elution mode. Mobile phase A was a 0.1% formic acid aqueous solution, and mobile phase B was acetonitrile. The elution procedures were 0–1.2 min, A:B = 70:30; 1.2–2.0 min, A:B = 10:90; and 2.0–3.5 min, A:B = 70:30. The detection and quantification were performed with a mass spectrometer (SCIEX, API 6500, Framingham, MA, USA), using the positive ion electrospray ionization method. The multiple reaction monitoring transitions for ketorolac tromethamine was *m*/*z* 256.1→105.0, and the internal standard (agomelatine) was 244.1→185.1. A standard curve was established for ketorolac tromethamine in the concentration range of 0.5–5000 ng/mL. The linearity was evaluated using weighted least-squares regression analysis, and the linearity was good in this concentration range (r^2^ > 0.995). The precision and accuracy of the assay met the experimental requirements.

#### 2.13.5. Calculation of Pharmacokinetic Parameters

Noncompartmental analysis of pharmacokinetic parameters in humans was performed using Phoenix WinNonlin 8.3 pharmacokinetic software (Certara, Princeton, NJ, USA). T_max_ and C_max_ were read from the blood concentration-time curve, and AUC_0-t_ was calculated using the linear trapezoidal rule. AUC_0-∞_ was the sum of AUC_0-t_ and C_t/_k_e_, where C_t_ is the last measurable blood concentration, k_e_ is the rate of decrease in blood concentration per unit time, and t_1/2_ is the time it takes for the blood concentration to drop by 50%.

### 2.14. Statistical Analysis

The software used for the statistics of this work was SPSS (IBM, Armonk, NY, USA), and the measures were expressed as the standard deviation of means (mean ± SD). If there was no statistical significance (*p* > 0.05), statistical analysis was performed using the one-way analysis of variance (ANOVA). If the ANOVA was statistically significant (*p* ≤ 0.05), a comparative analysis was performed using the LSD test (parametric method). If the variance was not equal (*p* ≤ 0.05), the Kruskal-Wallis test was performed. If the Kruskal-Wallis test was statistically significant (*p* ≤ 0.05), the comparative analysis was performed using Dunnett’s test (nonparametric method). Statistical results were tested at α = 0.05, where *p* ≤ 0.05 was considered statistically significant and *p* ≤ 0.01 indicated that the difference tested was highly significant.

## 3. Results and Discussion

### 3.1. Ideas for the Construction of Ketorolac Tromethamine Cataplasm

The principle for preparing the developed cataplasm was to use the cross-linking reaction between high-valent metal ions and water-soluble polymer materials. We used tartaric acid to free Al^3+^ from DAAA so that it would react with carboxyl functional groups in partially neutralized sodium polyacrylate. With the progress of the reaction, the viscosity of the mixture gradually increased, and the coating operation was carried out after the viscosity was appropriate. The purpose of adding EDTA-2Na was to chelate with free Al^3+^, thereby reducing the speed of the cross-linking reaction and ensuring sufficient time for coating. Our previous exploratory studies have shown that the ratio of polymer molecules, cross-linking agents, and cross-linking regulators are core factors for the successful cross-linking of cataplasm. Al^3+^ can react with the polymer to finally form a three-dimensional network structure so that the matrix can obtain appropriate strength. Divalent ions such as Ca^2+^, Zn+, and Mg^2+^ also have a cross-linking effect, but the matrix strength after complete cross-linking is poor and the cross-linking speed is too slow, so we did not use it in the final preparation method.

The prepared cataplasm had a high water content (about 40%), which ensured good skin affinity, and hydration also helped the transdermal diffusion of the drug [24]. To prevent water loss, sorbitol and PEG 400 acted as humectants, but high water content provided a favorable environment for the growth of microorganisms, so we added methylparaben and propylparaben as bacteriostatic agents [25]. To prevent the oxidation of ketorolac tromethamine, BHT was added as an antioxidant. Kaolin and titanium dioxide played the role of excipients, and povidone K90 ensured that the matrix had a high viscosity and had the effect of preventing the precipitation of drug crystals. 1,2-Propanediol was used as a penetration enhancer to improve the penetration performance of ketorolac tromethamine. Due to our use of l-menthol, the prepared cataplasm had a pleasant minty smell and a cooling sensation when applied to the skin, which helped to reduce the itching of the sore skin and improved patient compliance [26].

### 3.2. Characterization of Ketorolac Tromethamine Cataplasm

It can be observed from Appendix A that the paste of the cataplasm was consistent and possessed a great appearance. During the stirring process of the paste, menthol was easily volatilized to form air bubbles; thereby, a small number of small holes appeared on the surface without impairing patient use. No paste was left on the film once the release liner was removed, demonstrating the paste’s great cohesiveness. Ketorolac tromethamine’s microscopic shape was rod-like, and the cataplasm paste did not include such crystals as those shown in Appendix A. Despite the presence of some elliptical crystals in the visual field, which were likely made of kaolin or other insoluble materials, this showed that the matrix had entirely dissolved the active ingredient.

The surface of the cataplasm paste exhibited a polygonal mosaic structure at −20 °C, according to ESEM analysis. The paste’s moisture liquefied or sublimated as the temperature rose and the air pressure fell, causing pores to form on the paste’s surface and enlarge over time, the polygonal mosaic pattern to gradually fade away, and the paste to take on an irregular shape. As the temperature rose above 0 °C, the paste developed a cross-linked mesh structure (Appendix A, Figure 1a), appeared viscous, and contained some insoluble particles (Figure 1b).

Because the cataplasm was in a dosage form that was applied directly to the skin, its excellent adhesive properties were crucial. The ability of the cataplasm to swiftly cling to the contact surface was expressed with the term “tack”, which describes the ability of the cataplasm to attach to the surface of the object under very light pressure. The rolling-ball tack method and the loop tack test were used to investigate the tack of ketorolac tromethamine cataplasm, and the peel strength was used to determine the cataplasm’s ability to resist interfacial separation from the surface to which it had adhered under the action of appropriate pressure and time. The apparent adhesion characteristics of cataplasm with various drug contents were not significantly different, as shown in Table 1, indicating that slight variations in drug dosage have little effect on cataplasm adhesion.

In addition, the ability of the cataplasm to adhere appropriately to the application site under only slight pressure was largely due to the ability of the cataplasm to undergo cold flow at room temperature, which allowed the paste to flow rapidly and wet the contact surface, gaining a large contact area, thus fixing itself to the contact surface for mechanical anchoring. The occurrence of cold flow required the solid material to have suitable liquid properties and exhibit both elasticity and viscoelasticity, which could be characterized by rheological properties. The LVR of the sample was obtained from the amplitude scan test. The sample’s ability to resist flow and structural stability within a specific pressure range was indicated by the location and length of the LVR. As shown in Figure 1c, the LVR of the sample was 10-1225 Pa, indicating that the sample possessed good resistance to flow and pressure [27]. A fixed stress value (50 Pa) within the LVR was chosen, and the results of the frequency scan are shown in Figure 1d. The sample exhibited a rheological characteristic of G′ > G″ at different vibration frequencies, i.e., more “solid” than “liquid”, reflecting the elasticity of the sample as a gel matrix with a cross-linked reticular structure. The variation in strain over time for constant stress (50 Pa) was studied for the cataplasm. As shown in Figure 1e, the strain of the cataplasm increased rapidly to a plateau under stress, and after the stress was removed, the strain could not return to the zero-stress state. This creep-recovery characteristic showed similar features to the Kelvin-Voigt model [20], which also reflected the presence of a cross-linked structure within the cataplasm. According to the shear rate, from high to low, the trends in stress and viscosity of the cataplasm were investigated. The sample exhibited lower viscosity at high shear rates and higher viscosity at low shear rates, with the viscosity curves conforming to the Cross model, with r^2^ = 0.999, and exhibiting a shear-thinning characteristic (Figure 1f).

### 3.3. In Vitro Release Study

As illustrated in Figure 2a and Appendix A, the cumulative drug release amount of the cataplasm was dose-dependent. The release of the drug in three dose groups (0.5%, 1.0%, and 1.5%) was rapid and plateaued at approximately 6 h, indicating that the cross-linked mesh structure of cataplasm would not limit the release of ketorolac tromethamine. The fitting results of the drug release kinetic model (Table 2) showed that the release curve best fit the simplified Higuchi equation, indicating that the cataplasm belongs to the erodible matrix drug-delivery system. The diffusion behavior of the drug itself in the cataplasm was the rate-limiting step of release, which was consistent with the release mechanism of conventional semisolid preparations [28,29]. The correlation coefficients of the regression equations for the three dose groups (0.5%, 1.0%, and 1.5%) were 0.9994, 0.9996, and 0.9988, respectively (Table 3, Figure 2b), revealing that the release model fitting was highly reliable.

### 3.4. Screening of Skin Penetration Enhancers

An ideal transdermal enhancer should be nontoxic, not irritate the skin, not cause an allergic reaction, and considerably improve skin permeability [30]. As shown in Figure 3a, the transdermal penetration rates of four enhancers within 24 h were ranked as follows: 1,2-propanediol > isopropyl myristate > Plurol^®^ Oleique CC497 > Transcutol^®^, and the amount of drug retention in the skin of different penetration enhancer groups was consistent with the trend of the cumulative penetration amount (Figure 3b). 1,2-propanediol has good biocompatibility, low skin irritation, and has the greatest ability to help ketorolac tromethamine penetration relative to the other transdermal enhancers in this study. Additionally, 1,2-propanediol assisted in delaying the loss of water from the cataplasm as a humectant. Therefore, it was chosen as an ideal penetration enhancer in the cataplasm of ketorolac tromethamine.

### 3.5. In Vitro Skin Penetration Study

It was generally believed that the transdermal penetration behavior of drugs in preparations used on the skin surface belonged to passive diffusion, with the driving force being the differences in drug concentrations between the cataplasm and the skin. Strictly speaking, it was driven by the medicinal chemical potential energy gradient, which can be described using Fick’s first law [31] (Equation (4)).
(4)$$J=-D\frac{\partial c}{\partial x}$$
where *J* is the steady-state transdermal permeation rate per unit area of the drug, *D* is the transdermal permeability coefficient of the drug, and $\frac{\partial c}{\partial x}$ is the drug concentration gradient.

When the solubility of the drug in the cataplasm matrix was less than the saturation solubility, the value of *J* was linearly positively related to the drug concentration gradient. As the drug concentration in the cataplasm increased, both the drug concentration gradient between the inside and outside of the skin and the value of *J* were higher (Figure 3c, Table 4). The cumulative penetration ratio of the drug was about 15% for 24 h, indicating that ketorolac tromethamine could penetrate the skin of Bama minipigs effectively. The *T_lag_* of cataplasm in different dose groups was shorter than 2 h, suggesting that ketorolac tromethamine may enter tissues to exert anti-inflammatory and analgesic effects quickly. The results of the in vitro permeation experiments and in vitro release test showed that the cumulative permeation ratio (about 4.7%) was much lower than the release ratio (about 90%) at the same time (8 h point), indicating that the release behavior was not the rate-limiting step of the permeation behavior (Figure 3c, Appendix A).

### 3.6. Histological Evaluation

It can be observed from Figure 4a,e,i,j that the skin structure of the untreated group was dense and orderly, whereas the internal structure of the skin after 24 h of exposure to the drug-containing cataplasm was loose and disordered (Figure 4b–d,f–h,k,l). This indicates that the cataplasm weakened the barrier function of the stratum corneum by changing the skin structure and improving the skin’s permeability, thereby facilitating penetration of ketorolac tromethamine into the skin. The epidermis structure treated with different dose groups did not exhibit an obvious difference and remained generally intact, clarifying that although it changed the structure of the skin, the cataplasm caused no permanent damage to the skin when the dose was less than 1.5%.

### 3.7. In Vivo Pharmacodynamic Study in SD Rats

Figure 5a illustrates how the rise in the PT change rate was restrained in the other groups as compared to the control group. In other words, each treatment showed different degrees of analgesic effect. Rats in the oral, mid-dose (1.0%), and high-dose (1.5%) groups exhibited lower rates of change in their PTs, with no significant differences between each group. This result suggests that our cataplasm can provide analgesic effects comparable to those of oral administration.

We also investigated the anti-inflammatory effects of different dosages of cataplasm applied versus taken orally in rats (Figure 5b). The rats’ toe volumes rose noticeably after the modeling process. The anti-inflammatory effect study found similar results to the analgesic effect study: the cataplasms of mid-dose or high-dose groups were effective in reducing toe swelling in rats, and our formulation exhibited good anti-inflammatory effects.

As shown in Table 5 and Table 6, the deviation of the pharmacodynamic index was a bit high, which likely stemmed from the large coefficient of skin variation in SD rats and their differences in sensitivity to inflammatory responses [32]. The decrease in the PT and swelling after 4 h of treatment could be caused by two factors. First, the maintenance time of the carrageenan-induced inflammation model is limited, and some of the literature presents similar results [33]. Second, the menthol in the cataplasm prescription has some analgesic effect, which may also contribute to the improvement of the pharmacodynamic index [34].

In both trials, the mid-dose and high-dose cataplasm showed similar therapeutic effects, which was compatible with the ceiling effect of NSAIDs [35], suggesting that the analgesic effect might not increase with a higher dose after reaching a certain threshold.

### 3.8. In Vivo Pharmacokinetic Study in Humans

#### 3.8.1. Subjects

Eight healthy Chinese male subjects were successfully enrolled from twenty-one individuals for the clinical study, with no subjects withdrawn. Appendix A demonstrates the demographic characteristics of the subjects in this clinical trial.

#### 3.8.2. Pharmacokinetic Properties of Ketorolac Tromethamine Cataplasm and Tablet

As shown in Figure 6 and Table 7, the C_max_ of the cataplasm group was 4.10 ± 2.15 ng/mL, which was much lower than that of the tablet group (1231.84 ± 380.25 ng/mL). The blood concentration of topically administered ketorolac tromethamine cataplasm was more stable in humans relative to oral tablets, indicating that the rate of ketorolac tromethamine in the cataplasm was essentially constant through the skin, exhibiting a slow absorption behavior. The AUC_0-t_ and AUC_0-∞_ were 166.31 ± 85.28 h·ng/mL and 355.17 ± 144.96 h·ng/mL, respectively, for cataplasm and 4450.48 ± 1383.62 h·ng/mL and 4489.58 ± 1414.70 h·ng/mL, respectively, for tablets, and the plasma exposure of the drug was much lower for cataplasm than for tablets. The systemic exposure of ketorolac tromethamine mostly caused side effects such as gastrointestinal injury and liver damage; therefore, the safety risk of administration in the form of topical application of cataplasm is lower compared to tablets.

#### 3.8.3. Safety and Tolerability Assessments

No serious adverse events were reported in either period (Table 8). In the first period (cataplasm-treatment group), no adverse events were reported. In the second dosing period (tablet-treatment group), a total of five (62.5%) adverse events were reported, including one (12.5%) headache and four (50.0%) instances of elevated bilirubin. All adverse events recovered spontaneously after the end of the period and were considered to be related to the subject drug. The lower incidence of adverse events in subjects in the cataplasm-treatment group compared with the tablet-treatment group reflected, to some extent, that the topical application of ketorolac cataplasm can reduce systemic side effects in patients.

## 4. Conclusions

In our study, we prepared a cataplasm of ketorolac tromethamine and characterized its microstructure, mechanical properties, and rheological properties, which showed that the matrix is a viscous elastomer formed by a cross-linked mesh structure. In addition, the results of in vitro release tests of different doses of cataplasm showed that the release of the drug within the paste satisfied the Higuchi model and exhibited a diffusion behavior. Through in vitro permeation tests, 1,2-propanediol with an optimal effect was selected as the permeation enhancer. According to the results of pharmacodynamic experiments in rats, cataplasm in 1.0% and 1.5% doses provided anti-inflammatory and analgesic effects similar to those of oral administration, and there was no significant difference in therapeutic effects between the two doses of cataplasm. We also verified the safety of the formulation in the human pharmacokinetic study, wherein subjects using cataplasm presented lower plasma drug concentrations and fewer reported adverse events. Results from trials in animals and humans suggested that the 1.0% dose of cataplasm achieves a reduction in the risk of adverse events without loss of efficacy and that our cataplasm is a potential formulation for the treatment of inflammatory pain.

## Figures and Tables

**Figure 1 pharmaceutics-15-01405-f001:**
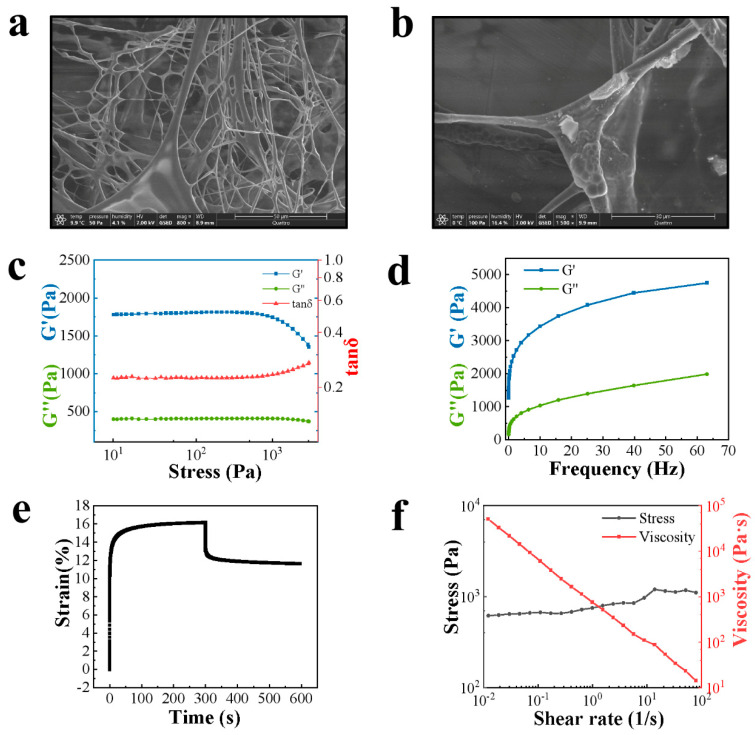
(**a**) ESEM observation of the cataplasm (800×). (**b**) ESEM observation of the cataplasm (1300×). (**c**) Variation in viscoelastic parameters with stress. (**d**) Variation in viscoelastic parameters with frequency. (**e**) Variation in strain in the creep-recovery test. (**f**) Variation in stress and viscosity with shear rate.

**Figure 2 pharmaceutics-15-01405-f002:**
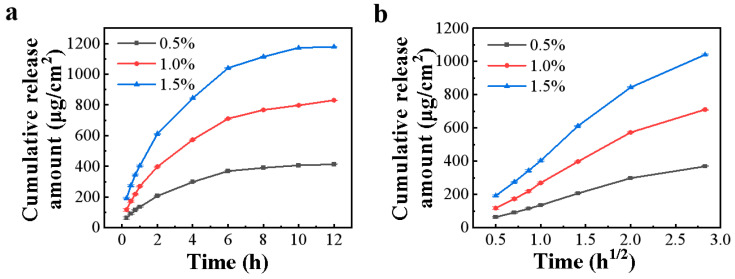
In vitro release rate of ketorolac tromethamine in different doses of cataplasm; (**a**) the variable in the horizontal coordinate is time (h); (**b**) the variable in the horizontal coordinate is time (h^1/2^); all values are expressed as mean ± SD, *n* = 6.

**Figure 3 pharmaceutics-15-01405-f003:**
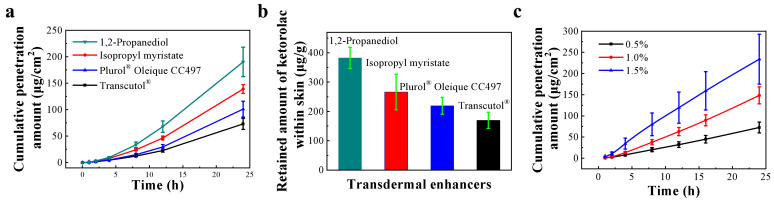
(**a**) Effects of skin penetration enhancer types on cumulative penetration of ketorolac tromethamine through Bama minipig isolated skin. (**b**) The amount of ketorolac tromethamine retained in the skin at 24 h after application of cataplasm containing different types of skin penetration enhancers. (**c**) In vitro penetration rate of ketorolac tromethamine with different doses of cataplasm. All values are expressed as mean ± SD, *n* = 6.

**Figure 4 pharmaceutics-15-01405-f004:**
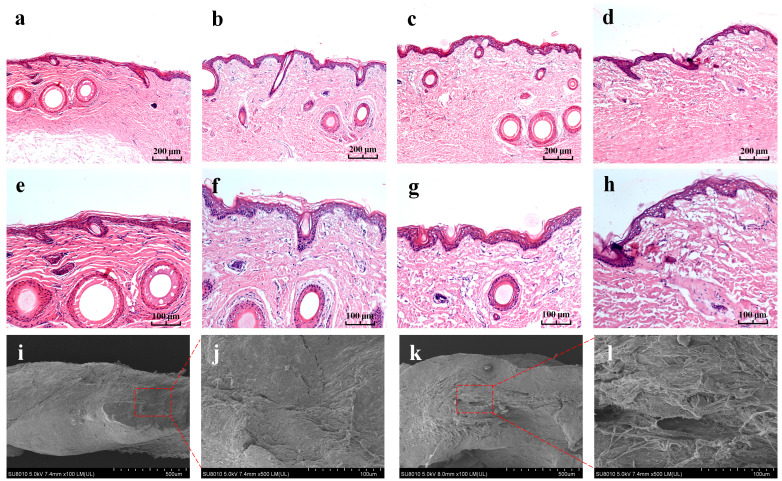
Ordinary optical micrographs (**a**–**h**) and SEM photos of the vertical section of the skin. At 100×: (**a**) the untreated skin and (**b**–**d**) skin treated with 0.5%, 1.0%, and 1.5% doses of cataplasm. At 200×: (**e**) the untreated skin and (**f**–**h**) skin treated with 0.5%, 1.0%, and 1.5% doses of cataplasm. (**i**) Untreated skin (100×); (**j**) untreated skin (500×); (**k**) ketorolac tromethamine-treated group (1.0%, 100×); and (**l**) ketorolac tromethamine-treated group (1.0%, 500×).

**Figure 5 pharmaceutics-15-01405-f005:**
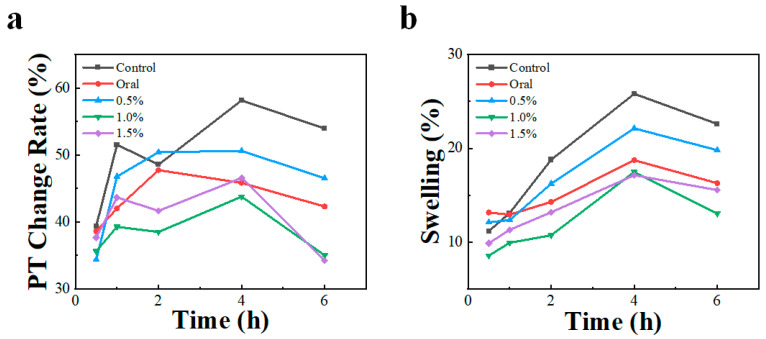
The rate of change in (**a**) PT and (**b**) swelling in model rats (n = 10) after administration of carrageenan-induced inflammation.

**Figure 6 pharmaceutics-15-01405-f006:**
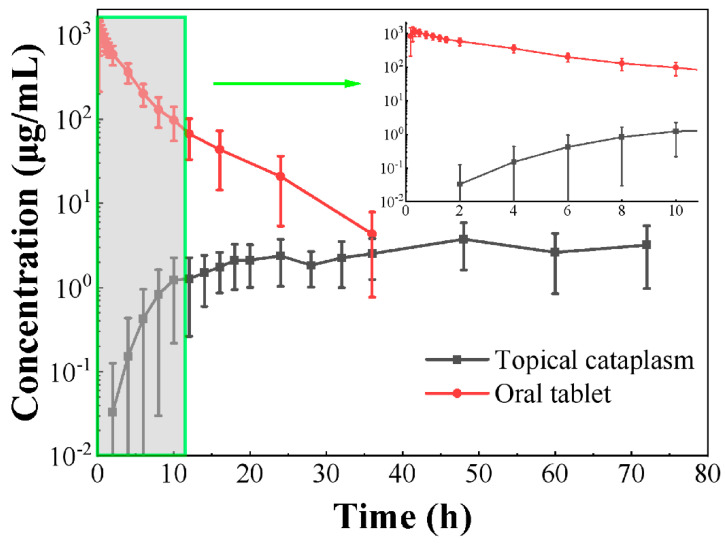
Plasma concentration-time curves of ketorolac tromethamine in healthy humans (fasting) after topical administration of ketorolac tromethamine cataplasm (100 mg) or oral administration of commercially available tablets (10 mg) (*n* = 8). Each point represents the mean ± SD.

**Table 1 pharmaceutics-15-01405-t001:** Apparent adhesion properties of the cataplasm with different drug contents.

Dose (%)	Tack		Peel Strength (N·cm^−1^)
	Rolling-Ball Tack Method (Ball Number)	Loop Tack Test (N)	
0.5	34	2.32 ± 0.24	0.0170 ± 0.0010
1.0	30	2.64 ± 0.27	0.0157 ± 0.0006
1.5	33	2.51 ± 0.18	0.0167 ± 0.0012

Peel strength values and values obtained using the loop tack test method are expressed as mean ± SD, *n* = 3.

**Table 2 pharmaceutics-15-01405-t002:** Kinetic model fitting results of release curves.

Kinetic Models	Fitted Equation	Dose (%)	r^2^
Zero-order	*Q_t_*/*Q_∞_* = *Kt*	0.5	0.9737
		1.0	0.9708
		1.5	0.9663
First-order	ln(1 − *Q_t_*/*Q_∞_*) = −*Kt*	0.5	0.9738
		1.0	0.9758
		1.5	0.9694
Higuchi	*Q_t_*/*Q_∞_* = *Kt^1/2^*	0.5	0.9994
		1.0	0.9996
		1.5	0.9988

Where *Q_t_* is the cumulative drug release amount at time *t*, *Q_∞_* is the drug release amount at time *t_∞_*, and *K* is the release rate.

**Table 3 pharmaceutics-15-01405-t003:** Release parameters fitted with the Higuchi equation.

Dose (%)	Fitted Curve	Release Rate (μg/cm^2^/h^1/2^)	r^2^
0.5	*Q* = 158.80*t^1/2^* − 20.01	158.80 ± 1.65	0.9994
1.0	*Q* = 306.64*t^1/2^* − 39.70	306.64 ± 2.40	0.9996
1.5	*Q* = 438.63*t^1/2^* − 29.96	438.63 ± 6.19	0.9988

All values are expressed as mean ± SD, *n* = 6.

**Table 4 pharmaceutics-15-01405-t004:** In vitro transdermal parameters of different doses of cataplasm.

Dose (%)	Fitted Curve	J (μg/cm^2^/h)	The Ratio of the CUMULATIVE Penetration Amount to the Dose at 24 h (%)	T_lag_ (h)
0.5	*Q* = 3.12*t* − 4.04	3.12 ± 0.07	16.9 ± 3.0	1.3 ± 0.2
1.0	*Q* = 6.44*t* − 10.73	6.44 ± 0.20	17.3 ± 2.4	1.7 ± 0.3
1.5	*Q* = 10.14*t* − 5.52	10.14 ± 0.18	13.6 ± 3.4	0.6 ± 0.2

All values are expressed as mean ± SD, *n* = 6.

**Table 5 pharmaceutics-15-01405-t005:** PT change rate of rats after administration.

Group	PT Change Rate (%)				
	0.5 h	1 h	2 h	4 h	6 h
Control	39.4 ± 14.2	51.5 ± 13.8	48.6 ± 10.7	58.2 ± 8.2	54.0 ± 12.0
Oral	38.6 ± 8.0	42.0 ± 7.3	47.8 ± 10.6	45.8 ± 10.6 **	42.3 ± 15.9
0.5%	34.4 ± 10.8	46.8 ± 4.8	50.4 ± 14.8	50.6 ± 10.8	46.5 ± 15.6
1.0%	35.6 ± 7.6	39.3 ± 12.6	38.5 ± 13.2	43.8 ± 16.5 *	35.0 ± 13.6 **
1.5%	37.7 ± 12.6	43.7 ± 7.3	41.7 ± 10.2	46.6 ± 14.8 *	34.3 ± 15.6 **

All values are expressed as mean ± SD, *n* = 10. Difference between each group and control group. * *p* < 0.05. ** *p* < 0.01.

**Table 6 pharmaceutics-15-01405-t006:** Rate of toe swelling in rats after administration.

Group	Swelling (%)				
	0.5 h	1 h	2 h	4 h	6 h
Control	11.2 ± 3.4	13.1 ± 3.6	18.8 ± 5.9	25.8 ± 10.1	22.6 ± 9.0
Oral	13.2 ± 2.7	12.9 ± 3.3	14.3 ± 6.1	18.7 ± 6.3	16.3 ± 8.1
0.5%	12.1 ± 4.1	12.3 ± 5.4	16.2 ± 5.2	22.1 ± 6.4	19.8 ± 4.1
1.0%	8.6 ± 4.5	9.9 ± 3.6	10.7 ± 5.9 **	17.5 ± 7.2 *	13.0 ± 8.1 *
1.5%	9.9 ± 3.0	11.3 ± 4.9	13.2 ± 3.5 *	17.1 ± 6.3 *	15.6 ± 6.8

All values are expressed as mean ± SD, *n* = 10. Difference between each group and the control group. * *p* < 0.05. ** *p* < 0.01.

**Table 7 pharmaceutics-15-01405-t007:** Human pharmacokinetic parameters (fasting) of topical ketorolac tromethamine cataplasm and commercially available oral tablet (*n* = 8).

Parameter	Topical Cataplasm	Oral Tablet
T_max_ (h)	48.0 (16.0, 72.0)	0.3 (0.2, 0.8)
C_max_ (ng/mL)	4.1 ± 2.2 (52.3)	1231.8 ± 380.2 (30.9)
AUC_0-t_ (h·ng/mL)	166.3 ± 85.3 (51.3)	4450.5 ± 1383.6 (31.1)
AUC_0-∞_ (h·ng/mL)	355.2 ± 145.0 (40.8)	4489.6 ± 1414.7 (31.5)

Where T_max_ is described as the median (min, max), and the remaining parameters are described as the mean ± SD (%RSD).

**Table 8 pharmaceutics-15-01405-t008:** Adverse events per period (*n* = 8).

Adverse Event	No. of Subjects (No. of Events)	
	Topical Cataplasm	Oral Tablet
Headache		1 (1)
Direct bilirubin elevated		1 (1)
Indirect bilirubin elevated		2 (2)
Total bilirubin elevated		1 (1)
Total	0 (0)	5 (5)

## Data Availability

The data presented in this study are available on request from the corresponding author.

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
