# Peer review of "Topical Delivery of Ketorolac Tromethamine via Cataplasm for Inflammatory Pain Therapy"

_pharmaceutics, 2023, doi:10.3390/pharmaceutics15051405_

Round 1
Reviewer 1 Report
In the submitted manuscript “Topical Delivery of Ketorolac Tromethamine via Cataplasm for Inflammatory Pain Therapy” prepared by Zhiyuan Hou et al., application of ketorolac tromethamine as a non-steroidal anti-inflammatory drug (NSAID) in topical form. For that purpose, a topical delivery system, cataplasm, based on a three-dimensional mesh structure formed by the cross-linking of dihydroxyaluminum aminoacetate (DAAA) and sodium polyacrylate was used. Cataplasm was characterized by microstructure, mechanical properties, and rheological properties. The in vitro release of different doses of cataplasm and in vitro permeation, as well as the safety of the formulation through human pharmacokinetic experiments have been investigated.Based on the presented results and conclusions, the submitted manuscript is acceptable with major revision, that authors should performed.
The main issues that should be resolved are the following:
1. The Introduction should be implemented state-of-the-art of delivery systems and release of ketorolac tromethamine.
2. 3.3. In vitro release study – : The experiments of In vitro release rate of ketorolac tromethamine in different doses of cataplasm ( fig 2 a and 2b) should be performed for longer time than t=2 h, a) for 24 h ; b) until reaching equilibrium. The obtained results should be discussed and compered with skin penetration experiments.
3. Results shown in Figure 5 a and b, should be reconsidered. Why so much deviations? The rate of change in (a) PT and (b) swelling abruptly decrease after 4h, which should be explained.
Author Response
Dear Reviewer:
On behalf of my co-authors, I appreciate you giving the positive and constructive comments and suggestions on our manuscript entitled “Topical Delivery of Ketorolac Tromethamine via Cataplasm for Inflammatory Pain Therapy” (ID: pharmaceutics-2293675).
We have studied your comments carefully and tried our best to revise our manuscript accordingly. Please see the attachment for specific responses.
Kind regards,
Jinsong Ding

Reviewer 2 Report
The reported work entitled "Topical Delivery of Ketorolac Tromethamine via Cataplasm for Inflammatory Pain 2 Therapy” is interesting. However, the manuscript can be accepted in Pharmaceutics after taking my concerns into account, as follows.
1. Many grammatical and typographical errors must be carefully corrected.
2. On a careful reading, the manuscript is a light and sequential description of references in the introduction section with insufficient contributions. In this regard, please try to presents, organizes and synthesizes the existing understanding about a topic, not to enumerate/ mention them.
3. There are many study about anti-inflammatory study, what is novelty in your study? Please discuss the novelty of your study in the introduction clearly.
Author Response

(The authors gave the same response as above.)

Reviewer 3 Report
The manuscript pharmaceutics-2293675 ''Topical Delivery of Ketorolac Tromethamine via Cataplasm for Inflammatory Pain Therapy'' by Zhiyuan Hou et al. describes the development of a topical delivery system of ketorolac by cataplasm based on dihydroxyaluminum aminoacetate and sodium polyacrylate. The resulting system was characterized using rheological and adhesion methods, and the morphology was demonstrated using SEM. Ex vivo and in vivo experiments confirmed the effectiveness of the obtained formulation. The manuscript is logical and well written. The paper will definitely be of interest to the readers of Pharmaceutics.
Questions and comments:
-
What is the mechanism of adhesion of the resulting cataplasm to the skin? How do the used methods (rolling ball-tack test and loop tack test) allow evaluating the adhesion to the skin?
-
Figure 2 - Clarify what (a) and (b) mean.
-
Figure 3 - The size of Figure 3 should be increased.
-
You have studied the activity of cataplasm in humans. Does this mean that the developed ketorolac-containing cataplasm is ready for commercial transfer?
Author Response

(The authors gave the same response as above.)

Round 2
Reviewer 1 Report
Since the authors poorly responded to my comments and practically didn’t accept essential suggestions, in my opinion, the manuscript is not acceptable for publication in the present form.
Author Response
Dear reviewer,
Thank you very much for your constructive comments, the logicality and readability of the manuscript have been substantially improved after revising it according to your suggestions. Please see the attachment for specific responses.
We hope that the quality of the revised manuscript will meet the standards for publication.
Best wishes,
Jinsong Ding

Reviewer 2 Report
Comments and Suggestions for Authors
I would like to thank the authors for adequately addressing all the comments.
I believe that the manuscript at its current form is in good shape for publication and that the quality of presentation, as well as the scientific soundness are significantly improved. I would suggest that the authors do a final revision and correct any minor spelling errors (subscripts/superscripts were necessary, space before units, etc.) before the final submission.
Author Response
Dear Reviewer,
Thank you very much for your positive comments, the quality of the manuscript has been substantially improved after revising it according to your suggestions. Minor errors in the manuscript have been rechecked and corrected in this round.
Best wishes,
Jinsong Ding